# Perception and coping mechanisms of patients with diabetes mellitus during the COVID-19 pandemic in Ibadan, Nigeria

Olajumoke Ololade Tunji-Adepoju[1], Obasanjo Afolabi Bolarinwa[2,3]*, Richard Gyan Aboagye[4], Williams O. Balogun[5]

**1** Department of Sociology, University of Wroclaw, Wroclaw, Poland, **2** Department of Demography and Population Studies, University of Witwatersrand, Johannesburg, South Africa, **3** Department of Public Health, School of Business and Health Studies, York St John University, London, United Kingdom, **4** Department of Family and Community Health, Fred N. Binka School of Public Health, University of Health and Allied Sciences, Hohoe, Ghana, **5** Department of Internal Medicine, College of Medicine, University of Ibadan & University College Hospital, Ibadan, Nigeria

* bolarinwaobasanjo@gmail.com

**Data Availability Statement:** The data utilised in this research contain potentially identifying or sensitive patient information; data are owned by the Institute for Advanced Medical Research and

## Abstract

### Background

The 2019 coronavirus disease (COVID-19) ushered in a period of fear and uncertainty, resulting in structural instability across the globe. Vulnerable individuals, such as patients with diabetes mellitus, are predispose to have adverse effects and complications of COVID-19 when infected. We explored the perception of diabetes mellitus patients during the COVID-19 pandemic and their coping mechanisms at the University College Hospital, Ibadan.

### Methods

We employed an exploratory qualitative study design to explore diabetes mellitus patients' perceptions and coping mechanisms during the COVID-19 pandemic. A purposive sampling technique was used to recruit 32 participants (2 health professionals and 30 diabetes mellitus patients). In-depth interviews were used to collect the data from the participants. All the recorded audio data were transcribed verbatim and exported to NVivo software for thematic data analyses.

### Results

Most diabetes mellitus patients were not fearful of the pandemic but were optimistic that it would not affect their health. Mechanisms such as the usage of herbal medicines and adherence to COVID-19 precautionary measures were noticed among patients. The study also revealed that the hospital's coping mechanism during the COVID-19 pandemic include prolonged appointments, limiting the number of patients attended per clinic day, and the provision of telehealth service. Patients in our study utilised negative coping mechanisms such as reduced drug dosages, subscriptions to cheaper drug brands, and reliance on religious institutions rather than a clinic for health instructions.

Training (IAMRAT). Please get in touch with IAMRAT via cfalade@comui.edu.ng or the first author for data request and access.

**Funding:** The author(s) received no specific funding for this work.

**Competing interests:** The authors have declared that no competing interests exist.

**Abbreviations:** COVID-19, Coronavirus Disease 2019; IDI, In-depth Interview; KII, Key Informant Interview; NCD, Non-Communicable Diseases; NCDC, Nigeria Centre for Disease Control and Prevention; WHO, World Health Organization.

## Conclusions

The study has shown that diabetes mellitus patients were not fearful of the COVID-19 pandemic. The utilisation of telehealth, encouragement of daily monitoring of sugar levels, provision of avenues for a medication review, and adherence to the safety protocols were coping mechanisms employed by the health system and diabetes mellitus patients. We recommend that the government and other healthcare stakeholders reinforce the resilience of diabetes mellitus patients by alleviating their health burdens during the pandemic. This could be done by subsidizing the prices of drugs, tests, and consultation fees for patients with diabetes mellitus. Also, more efforts should be made to elevate the health system through the reduction in waiting and appointment times in the diabetes clinic and employing more health personnel in the clinic.

## Background

Diabetes Mellitus has been an important global health concern even prior to the outbreak of the 2019 coronavirus disease (COVID-19). Diabetes mellitus is a chronic health condition characterized by high blood glucose due to either the inability of the pancreas to produce adequate insulin or the body's resistance to available insulin [1]. Poorly managed diabetes mellitus can result in long-term complications such as amputation, cardiovascular disease, vision impairment, and renal disease [1]. The prevalence of diabetes mellitus is increasing across the globe. Between 1980 and 2014, the number of patients with diabetes mellitus increased from 108 million to 422 million [2] and over 500 million in 2020 [3, 4]. Diabetes mellitus is projected to be the seventh leading cause of death worldwide by 2030 [2]. The burden of diabetes is greatly increasing in sub-Saharan Africa, and Nigeria has the largest share of this burden [5, 6]. Evidence suggests that an estimated 11 million Nigerians have diabetes [7]. This implies that more than five percent of the Nigerian population is living with the disease, and it is a major cause of death among many Nigerians below 70 years old [8].

Research has shown that the presence of excess blood glucose, also known as hyperglycemia in diabetes patients, makes them susceptible to COVID-19 [9, 10]. In the same vein, a study reported that patients with diabetes are two times more likely to develop severe conditions or die from COVID-19, while people living with uncontrolled diabetes are about 13 times more likely to die from the virus [9]. This has instilled fear in many patients with diabetes in Nigeria [10]. Hence, the fact that patients with diabetes need extra health care and attention during the pandemic remains undisputable [11]. Measures to prevent and control Non-Communicable Diseases (NCDs), such as diabetes, in developing countries, including Nigeria, have not been effectively implemented. The COVID-19 pandemic worsened the accumulated effects of Nigeria's failure to improve the healthcare structure and system, particularly the care of people with NCDs. The outbreak of COVID-19 in Nigeria on the 27th of February 2020 [12] also caused dysfunctionality and complications in the Nigerian social structures, requiring sporadic responses.

Predictably, the response to COVID-19 in Nigeria infringed on treating other diseases, including diabetes mellitus [13]. For instance, many public hospitals were converted to COVID-19 treatment centers resulting in restricted access of other patients to medical care [5, 14]. Also, the World Health Organization (WHO) stipulated in its report that the COVID-19 pandemic negatively impacted the equity regarding essential health service delivery in its

member states [4, 17]. The report further revealed that more than 50% of its member states experienced disruption in delivering health care services on NCDs, cancer, and mental health disorders. Some of the reasons given for this disruption include shortage of staff due to their transfer to COVID-19 care centers, lack of public transport, and cancellation of planned appointments, while the reasons recorded in 20% of the member states were: the shortage of medicines, diagnostics, and other technologies [4]. Due to the various forms of hardship or discomfort caused by the pandemic, vulnerable individuals, such as people with diabetes mellitus, are likely to perceive the period differently than healthy people. Patients with diabetes are likely to have a negative perception of the COVID-19 condition, which could influence the adoption of harmful coping mechanisms that could further hamper their health outcomes. Although several perception studies have been done on COVID-19 generally in Nigeria, such as that of Osahon and Memudu [15] on the perception and healthy attitudes of Nigerians to COVID-19, very little data is available about the perception of diabetes patients of the COVID-19 pandemic. This study explored the perceptions of patients with diabetes mellitus during the pandemic and their coping mechanisms during the period. This is important to illuminate the experiences of patients with diabetes mellitus during the pandemic, and the findings could be beneficial to the government and healthcare stakeholders in formulating policies to help improve the health outcomes of patients with diabetes during the pandemic.

## Methods

### Study design

We adopted an explorative qualitative research approach [16]. This approach is suitable for understanding the perception of diabetes mellitus patients at University College Hospital and the coping mechanisms during the COVID-19 pandemic. This qualitative research approach also guide the collection of in-depth data from the participants in their natural form [16].

### Study setting

The study was conducted in the University College Hospital, Ibadan. University College Hospital is a tertiary health center that serves Oyo state and neighbouring states where patients with diabetes, including those with suspected COVID-19 cases, are seen. According to statistics from the Nigeria Centre for Disease Control and Prevention (NCDC) [17], Oyo State is among the COVID-19 most infected states in Nigeria, and Ibadan is its capital city. The University College Hospital was founded in November 1952, located at Oritamefa in the Ibadan North Local Government Area. It is the first teaching hospital in Nigeria to provide in-patient and outpatient health care services. It receives referrals from southwestern and other parts of Nigeria and outside the country. The Diabetes clinic runs every Monday, and an average of 50–70 patients visit the clinic daily.

### Participant sampling and sample size

Participants in this study were registered diabetes patients who had been visiting the outpatient diabetes clinic at University College Hospital before the pandemic and are still attending the clinic during the pandemic. We used a purposive sampling technique to recruit 30 participants for the study, which was determined using the data saturation method. These participants constituted those who experienced the phenomena under study. Patients who were living with diabetes mellitus disease prior to the pandemic (at least more than a year) and aged 18 years and above were included. However, those accessible during the data collection period were included. The rationale was that this category of patients with diabetes mellitus could tell the

difference between their experiences in the clinic before and during the pandemic. Being coherent, healthy, gave informed consent were other inclusion criteria for the study. Patients who did not consent to the study or were no longer interested in participating even while in the middle of it were excluded from the study. Patients diagnosed with diabetes after the outbreak of COVID-19 and/or who did not attend the diabetes clinic during the study period were not included. Also, patients with acute or chronic debilitating comorbidities were not included in the study.

## Data collection procedure

Four (4) trained research assistants were used for the data collection. The research assistants were trained anthropologists, who the authors also trained in areas consisting of the consenting process, interviews, asking probing questions, and recording. The authors developed an interview guide as the data collection tool. Data on the participants' sociodemographic characteristics and the study objectives were collected after obtaining consent. The probe questions were structured to capture responses to the participants' perceptions of the COVID-19 pandemic. It also covered responses to the coping mechanisms adopted by diabetes mellitus patients during the pandemic. Prior to the data collection, we informed the participants about any possible discomfort, benefits, and compensation associated with the study. Interviews were conducted once the participants agreed to a date, time, and place of convenience to participants. The participants were approached for recruitment at the end of the medical appointments, and those who consented were interviewed at the premises of the hospital. The data collection lasted for an average of 60 minutes per participant. The participants were compensated with an equivalent gift of three US dollars ($3) at the end of the interview.

About 15 interviews were conducted each week for two weeks by one of the authors with four trained anthropologists who had qualitative fieldwork experiences. The anthropologist helped in administering the interview guide. Interviews were conducted with strict adherence to the COVID-19 precautionary measures. The data collection/interviews were done between the 15th to 22nd of March 2021 using a pretested interviewer guide. The participants were recruited using a purposive sampling technique. All the interviews were tape-recorded, and field notes were taken and utilised during the transcription and analysis. Data transcription was carried out after every fieldwork, and this helped in identifying questions that may have been left unanswered during the interview or those needing further probing, as well as identifying the point of saturation where no further interviews were conducted. The research assistants helped with field notes, tape recording and data analysis.

## Data analysis

All the audio data was transcribed verbatim on the same day the data was collected. After the transcription by the research assistants, After the transcription of the data, the transcripts were vetted and proofread by OOTA, OAB and WOB. Later, the transcripts were made accessible to OAB, who performed the initial independent thematic analysis [18]. Using the 'nodes' function in NVivo-12 software, where codes were assigned to the text data from the transcripts [19]. During the analysis, all similar recurring codes were categorised to generate themes and, subsequently, sub-themes [18]. The extracts and quotes from the themes and sub-themes generated were used to support the results of the study. All the authors approved the extracts and quotes. A pretest of the interview guide was done with two (2) potential respondents (male and female) among those who came for medical appointments prior to the actual commencement of the main data collection. The interview guide pretest results show accurate consistency, but the results were not included in the main study.

### Rigour and trustworthiness

In every qualitative study, credibility and trustworthiness measures are critical. In achieving this, we allowed two research assistants with experience in qualitative analysis to transcribe and analyze tape-recorded interviews separately. The two research assistants' themes and sub-themes, as well as the authors, were compared to ascertain their consistency. To strengthen the credibility of the results, direct quotations and precise summaries of participant remarks were used. A week after the transcription and preliminary data analysis, we conducted member-checking with three of the participants to demonstrate trustworthiness. This allowed the participants to attest that the transcripts accurately captured the content of the interviews. Nobody offered changes or voiced complaints about the interviews' calibre or content in terms of clearly expressing their viewpoints. The participants' nonverbal cues, their concerns, and the interviewers' observations were all documented in the field notes that were taken following each interview and consulted throughout the research. The authors who carried out the interviews are qualified healthcare researchers with expertise in conducting IDIs.

### Ethical issues

Ethical clearance was sought from the Ibadan/University College Hospital Ethics Committee (UI/UCH EC) with approval number UI/EC/21/0064. In this study, we complied with all the ethical guidelines pertaining to using human participants and peculiar to qualitative studies. We anonymized all the transcripts and audio files by giving them pseudonyms to remove any personal information that may be used to identify the study participants. The participants in the study were given an information sheet that included information on the objectives, methods, potential risks and advantages, compensations, who to contact, and an affirmation of confidentiality, privacy, and autonomy. The participants gave written consent by signing the consent form for participating, and for recording the interviews. Later, the participants' signed informed consent was requested. This demonstrated that they had read and understood the terms of reference before deciding to participate freely in our study. We also encrypted a passcode and locked the audio files and transcripts to prevent unauthorised individuals from accessing the material.

## Results

In-depth interviews were held with thirty patients with diabetes mellitus in the outpatient ward of the endocrinology clinic. Each interview lasted for about an hour.

### Demographic characteristics of participants

A total of thirty participants were recruited for the study, consisting of approximately two-thirds females and one-third males. Most participants were elderly, with the oldest being an 84-year-old female and the youngest a 21-year-old male. More than half were Christians, with the remainder being Muslims. All males, except the youngest, were married. Among the females, most were married, with two widows, one single and one separated.

Over half of the participants had tertiary education, a few had secondary education, and a small number had primary education. Only a few females had no formal education, whereas all males had some level of formal education.

Most males were retired, with three employed and one unemployed (the youngest male). Among females, more than two-thirds were employed, three were unemployed, and two were retired. Participants' socio-economic status ranged from average to low.

Concerning diabetes type awareness, seven out of ten female participants did not know their type, while all but one male participant was aware. Most participants had type 2 diabetes, except for one male and one female in their twenties who had type 1 diabetes.

## Emerging themes from the study

Table 1 presents the major themes and sub-themes that emerged from the study. While analyzing data, two key themes emerged: The perception of Diabetes Patients during the COVID-19 Pandemic and the coping mechanisms employed by diabetes mellitus patients during the pandemic.

## Perception of diabetes mellitus patients during the COVID-19 pandemic

This theme contains three sub-themes (clinic appointments, effects of the COVID-19 pandemic on diabetes patients, and adherence to COVID-19 protocols).

**Clinic appointments.** Data from the key informant interviews revealed that the pandemic was perceived as a period that has negatively affected appointments, clinic attendance, and the management of diabetes in the clinic. A participant stated that

> Yes, like I just told you, it has affected it a lot because I have been coming for months now, and I have not been able to see the doctor (IDI, Participant 10, Female, 45 years).

> The pandemic has affected my clinic attendance because, presently, I only come to the clinic when the doctor gives me appointments. Prior to the pandemic, asides from my appointment dates, I come to the clinic every first Monday of the month because we diabetes patients usually hold meetings in the diabetes association office; however, since the

**Table 1. Summary of emergent themes.**

| Theme | Sub-theme | Code |
|---|---|---|
| Perception of diabetes mellitus patients on the COVID-19 pandemic | Clinical appointment | Effect of COVID-19 on diabetes patients' appointment |
| | Effect of the COVID-19 pandemic | ✓ Vulnerable groups during the pandemic of diabetes patients<br>✓ Fear |
| | Adherence to COVID-19 protocols | ✓ Physical presentation of patients only when necessary<br>✓ Allowance of physical presentation of patients with strict adherence to COVID-19 protocols |
| Coping mechanisms adopted by the patients with diabetes mellitus | Mechanisms adopted by the clinic | ✓ Adoption of an online database<br>✓ Provision of avenues to review prescriptions<br>✓ Encouragement of daily monitoring of blood sugar<br>✓ Telehealth<br>✓ Adherence to COVID-19 protocols |
| | Personal coping mechanisms adopted by diabetes mellitus patients | ✓ Adherence to COVID-19 protocols<br>✓ Media advice |
| | Reasons for low subscription to telehealth | ✓ Lack of need to utilize it<br>✓ Non awareness of the service |
| | Dangerous mechanisms adopted by diabetes mellitus patients | ✓ Usage of traditional medicines<br>✓ Usage of old prescriptions<br>✓ Inconsistency in clinic attendance |
| | Suggested coping mechanisms | ✓ Provision of subsidized drugs and services<br>✓ Reduction of waiting time<br>✓ Continuation of diabetes association meeting<br>✓ Provision of more call lines staff<br>✓ Improvement of services provided by staff<br>✓ Improvement of facilities |

emergence of the pandemic, we have stopped holding the meeting (IDI, Participant 9, Male, 21 years)

**Effects of the COVID-19 pandemic on patients with diabetes mellitus.** This sub-theme has ten codes, which include the vulnerable groups during the pandemic. The participants revealed a clear understanding of the fact that diabetes patients are vulnerable during the pandemic. A number of the patients demonstrated a clear knowledge of the category of people who are vulnerable to COVID-19 infection. They perceived this category of people as the aged, those who defy preventive measures, and people with comorbidities. However, some of the participants were unaware of who could be susceptible to the virus.

Some of the diabetic patients who were not observing the COVID-19 precautionary measures are vulnerable to the virus, as encapsulated in the excerpt below:

*Anybody can contract it, especially those who fail to wash their hands and follow the preventive measures (IDI, Participant 2, Male, 75 years).*

*Those who fail to adhere to the necessary precautions, those who talk with their whole mouth open without using their nose masks and those who stay too close to other people (IDI, Participant 9, Male, 21 years).*

The elderly/people with underlying diseases Yes, people that are already sick before and the elderly ones (IDI, Participant 10, female, 45 years).

*It cannot have any negative impact on diabetes patients if they take their medications regularly and follow necessary precautions (IDI, Participant 9, Male 21 years).*

Some participants, however, did not know who could get infected with the virus, as evidenced in the below excerpt:

*I can't say (IDI, Participant 1, female, 74 years).*

*Only God knows (IDI, Participant 4, female, 84years)*

They posited that the COVID-19 virus could have severe impacts on them. The opinion of one participant in this regard is captured in the excerpt below:

*The diabetes patient could get infected and the person might not get cured of it. It could even kill such a person (IDI, Participant 8, Female, 52years).*

A few others were of the opinion that COVID-19 cannot have any deadly implications on patients with diabetes since one can be treated if infected.

*The impact it can have is for one to get infected, and even if one gets infected, since it can be treated. I don't think it can bring about any deadly impact except for someone whose time is up on this planet earth already (IDI, Participant 12, Female 28 years).*

Patients with diabetes are not vulnerable in as much as they follow their diabetes regimen:

*I don't think there should be any effects if they use their drugs regularly, exercise as well and take away the fear of contracting*

*the disease (IDI, Participant 2, Male 75 years).*

The in-depth interview further revealed that most of the participants were not afraid of the COVID-19 virus. The submission of a participant is clear on this, as evident in the statement below:

*Ehn…It did not make me scared because I firstly did not believe it was real but as time went on and I started hearing that people were really contracting it, it was then I believed that COVID-19 is real. However, till now, I have not seen anyone who has contracted it in my vicinity. As such, that is one of the reasons I did not really get scared about it (IDI, Participant 12, Female 28years).*

Patients with diabetes mellitus noted that they were not fearful of the pandemic because they trusted in the Supreme Being, their object of worship (God). The results further showed that a couple of patients with diabetes mellitus were not afraid during the pandemic due to their trust in their medications, the quality of health care provided in the University College Hospital diabetes clinic and their proper adherence to the COVID-19 precautionary measures. The excerpts below capture this:

*I'm unruffled. I know we have problems all over the world but I'm a Christian and I'm unruffled (IDI, Participant 2, Male 75 years).*

*I was not afraid, I don't believe in it. I have Jesus (IDI, Participant 21, Male 65 years).*

On the contrary, a few participants perceived that the COVID-19 pandemic was a fearful one, and they perceived it to be the end of their life. This is evident in the excerpts below:
A fearful period

*I was thinking I was going to die (IDI, Participant 14, Female 30years).*

*I was very scared, but God's grace is sufficient (IDI, Participant 1, Male 74 years).*

**Adherence to COVID-19 protocols.** This sub-theme encompassed the physical presentation of patients only when necessary and allowance of physical presentation of patients with strict adherence to COVID-19 protocols.
Findings from the interviews revealed that the clinic avoided unnecessary physical presentation of diabetes mellitus patients during the pandemic by prolonging appointments. Patients were advised to call doctors on the clinic lines except when there is a pressing health concern that requires a physical presentation of the patient in the clinic. Also, the opinion of a participant revealed that patients are informed to adhere to the COVID-19 precautionary measures whenever they need to be physically present in the clinic. These opinions are encapsulated in the opinions below:
Adherence to the COVID-19 safety protocols through prolonged appointments

*They did well before the pandemic, and they answered us on our appointed dates, but since the pandemic started, I'm just coming to see the doctor, it's over a year already. I've been coming in the previous weeks but I was sent back home saying that only when the doctor calls me is when I can come. All these were not there before the pandemic; they kept to their appointments then ((IDI, Participant 18, Female 80 years).*

*Yes, I am faithful but here, they are not faithful. The staff, attendants and the records officials are not faithful. Sometimes, when I come on my appointment dates, on getting here, they would tell me they have rescheduled my appointment because of COVID-19 (IDI, Participant 10, Female 45 years).*

*I didn't visit the hospital when the pandemic began, but when the hospital was open, I came, and we were told to use our drugs, and I haven't been able to see the doctor again. It's been a year I saw a doctor; I just saw the doctor on the first of March this year since all these while. Whenever I have appointments, I use to come, but University College Hospital is not helping matters cause most times I will come, they will say they've re-scheduled my appointment many times (IDI, Participant 22, Female, 72 years).*

One of the diabetes mellitus patients indicated that the *COVID-19* precautionary measures were mandatory for all patients in the clinic.

*They make sure we use our nose mask, they do temperature checks before you come in, even at the gate, they tell you to put your mask on (IDI, Participants 20, Male 75 years).*

## Coping mechanisms adopted by patients with diabetes mellitus

Coping mechanisms emerged as the second theme. The sub-themes were mechanisms adopted by the clinic, coping mechanisms adopted by the patients with diabetes mellitus, reasons for low subscription to telehealth, and dangerous coping mechanisms used by the patients with diabetes mellitus.

**Mechanisms adopted by the clinic.** Under this sub-theme, the patients with diabetes mellitus mentioned that the coping mechanisms put in place at the diabetes clinic include the adoption of an online database, provision of avenues to review prescriptions, encouragement of daily monitoring of blood sugar, telehealth, and awareness creation. The following narrative quotes to support this sub-theme;

*Now that they're using technology, things are improving, so little by little, I hope they will do better (IDI, Participants 11, Female 65 years).*

They treat me well each time I come. I am always instructed to get an exercise book to write my blood sugar level when I check it every morning and night every day. Once I come to the clinic, I do show the doctor the book. If it is normal, the doctor will not increase my medications, but if it is high, my medications will be increased. That is the way I am attended to (IDI, Participant 12, Female 28 years)

**Telehealth.** Participants shared their experiences with the communication services provided by the clinic:

They have call center, they introduced a call center that patients can call and speak to the doctors and nurses, errm I think it's not 24 hours, but its during working hours (IDI, Participants 17, Male 77 years).

I am only aware of the fact that I got a message from the clinic during the early period of the pandemic that I should call some numbers in case an emergency issue arises concerning my health (IDI, Participant 8, Female 52 years).

**Coping mechanisms adopted by patients with diabetes mellitus.** The patients utilized positive coping mechanisms such as adherence to preventive measures and reliance on the

media for COVID-19 updates. This is captured in the responses below by a number of the participants:

Adherence to the COVID-19 precautionary measures was one of the coping mechanisms the patients with diabetes mellitus adopted in preventing themselves from COVID-19 infection.

*They make sure we use our nose mask, they do temperature check before you come in, even at the gate they tell you to put your mask on (IDI, Participants 20, Male 75 years).*

*I simply follow the laid preventive measures and I keep myself clean in the house' (IDI, Participant 4, Female 84 years).*

*I was just doing what I was supposed to do and didn't do what i wasn't supposed to. For example, I have been avoiding crowded places. I have not attended Jumat service since the pandemic began rather, my family and I observe our prayers together at home (IDI, Participant 3, Male 63 years).*

A couple of the patients with diabetes mellitus relied on the frequent updates on COVID-19 from the mass media as a coping mechanism against the pandemic.

*I listened to health programs on the radio and I adhered to their health advices and I also took precautions. It was the precautionary messages and COVID-19 jingles delivered by newscasters on the radio I listened and adhered to (IDI, Participants 8, Female, 52 years).*

A few of the participants averred that though they were aware of the telehealth service provided by the clinic, they did not utilize it as a coping mechanism. This is vivid in the opinion of some participants.

*During the COVID I received text messages inviting me to come for my check up. But I didn't come ooo!. But they're trying (IDI Participants 11, Female 65 years).*

*I didn't because there was no reason for me to call them (IDI, Participants 3, Male 63 years).*

One of the patients opined that they were not aware of the telehealth service provided by the clinic. The provision of the telehealth service was helpful as it came as a timely intervention for some patients who utilized it. This is evident in the response of one of the participants, who opined thus:

*Yes during the pandemic when we wanted to see the doctor, we were told to call him on the phone, when I called the doctor I was told the drugs to buy (IDI, Participants 30, Female 44 years).*

**Reasons for low subscription to telehealth.** The findings of the study revealed that despite the provision of telehealth service as a coping mechanism, there are however some limitations to its use among patients with diabetes mellitus in University College Hospital.

One of the diabetes mellitus patients opined that he did not use it because a need to utilize the service did not arise. The excerpt below encapsulates this.

I didn't because there was no reason for me to call them (IDI, Participants 3, Male 63 years)

Also, a level of full awareness regarding telehealth has not been reached among diabetes mellitus patients in the clinic; as such, a few patients did not utilize the service because they were unaware of it.

No, I did not. I wasn't aware of that service (IDI, Participants 4, Female 84 years).

No, I am not aware of it (IDI, Participants 12, Female 28 years).

**Dangerous mechanisms employed by patients with diabetes mellitus in the clinic.** The study revealed the employment of negative mechanisms by diabetes mellitus patients in the clinic. This is evident in the excerpts below:

The use of herbal medicines

*I used some traditional herbs to also compliment my medications. I drank the juice from boiled mango leaves and ginger. I also took my injections. I stayed indoors and maintain social distancing the few times I go out. Most times even if I feel like going out, my parents will not allow me go out because they know that I am more vulnerable to COVID-19* (IDI, Participants 12, female, 28 years).

*I do a lot of steaming, herbal steaming, I cook dongoyaro leaves and the bark, we boil it, and we drink on the first day and subsequent days we just steam. Everybody in my house steams, we use menthol, we add menthol to it and we take a lot of supplements, vitamin C and D, that's all we've done so far* (IDI, Participants 17, Male 77 years).

Usage of old prescriptions

I continued buying the drug that was prescribed to me since the last time I came to the clinic. By the time my health deteriorated recently and I have been coming to the clinic, the prescription was changed, but I was already used to my old prescription (IDI, Participant 12, Female, 28 years).

Inconsistency in clinic attendance

Many at times, most of us do not come to the clinic because we do not have money (IDI, Participant 9, Male 21 years).

**Suggested coping mechanisms for the management of the University college hospital clinic and the government.** The patients with diabetes mellitus gave some recommendations on how the management of the University College Hospital Diabetes Clinic and the government can assist in easing the stressful conditions of diabetes mellitus patients during the COVID-19 pandemic. They are captured in the excerpts below:

Some participants want their medications to be subsidized and be easy to get.

*What I think can be done is. . ..You see, there is a saying that someone who has diabetes and does not want to die early needs money because it is not easy if I would be honest with you. I think what they can do is if they can help us subsidize our drugs. If we are given free drugs once in a while even if it is one drug, it would go a long way. What can be done is if they can help us that way once in a while then if they can help us find means through which our medications will not be scarce to get and they should help us such that it would not be too expensive than what we can bear. See. . . if it is not too expensive, there is nobody that does not want to use their medications and be healthy but the inability to afford the drugs makes one not to be faithful with the medications. Those are the ways I think they can help us*(IDI, Participant 12, Female 28 years).

They can make the drugs cheaper, some can't afford it and it led to their death(IDI, Participant 18, Female 52years)

*They should provide free drugs for us. Many at times, most of us do not come to the clinic because we do not have money(IDI, Participant 9, Male 21 years).*

Some patients with diabetes mellitus mentioned that the consultation fees should be reduced and this is captured in the narratives below.

*One thing they can do is to reduce the consultation fees. Sometimes after paying the money, there won't be anything left anymore. And sometimes, if we pay and the doctor is not around, the hospital won't refund the money; that one is gone (IDI, Participants 30, Female 44 years).*

*We are not equal and not all buoyant. The consultation fee can be reduced and some things can be given for free, even if it is a few drugs (IDI, Participants 13, Female 49 years).*

Reduction of waiting time was one of the recommendations suggested by patients with diabetes for the management of University College Hospital. This is supported by the quote below;

*Ah the major thing is that they should answer us in time. I understand that they will first do ward round but immediately they come back they should answer us (IDI, Participants 11, Female 65 years).*

One of the participants opined that there should be continuous or regular meetings which can be held in an open space within the clinic in other for them to be updated on their next line of action such as receiving medications, medication review, and adherence to COVID-19 protocols. The except below summarises this assertion.

*We have an association but* University College Hospital *is not allowing us to hold meetings now, and its affecting a lot, through the association, we have had numerous lectures from physiotherapist and dieticians, to teach us more about our condition but our venue is small and due to COVID-19 regulations, those meetings can't hold but we have pleaded with the management to allow us to do it outside in the open. Because the absence of those meetings is affecting some of us who have no clue as to go about something regarding our ailment (IDI, Participants 26, Female 68 years).*

Other suggested coping mechanisms include: opening more call lines to ensure efficient telehealth services, ensuring patients are treated politely in the clinic, and provision of hygienic toilet facilities.

## Discussion

This study explored the perception and coping mechanisms employed by patients with diabetes mellitus during the COVID-19 pandemic. We found that most participants were not fearful due to the pandemic; rather, they were optimistic while they played their part in ensuring they were safe. This is consistent with the findings of a study conducted in India [20], which revealed that most of the participants in their study were not so anxious about the pandemic but were rather optimistic. Also, the participants in this study demonstrated a clear understanding of those vulnerable to COVID-19, as some posited that anybody who defies the COVID-19 precautionary measures, people with underlying diseases such as diabetes, are

susceptible to the virus. This attests to the supposition that COVID-19 sensitization and training were done in communities and health facilities. Also, surveillance mechanisms were improved in communities in Nigeria [20].

Literature has revealed that some health facilities had to shut down the entire clinic, including diabetes outpatients, to protect the health care providers and patients from contracting the virus [21]. Similarly, our findings show that the vulnerable nature of the patients informed why all appointments in the University College Hospital diabetes clinic were cancelled during the early period of the pandemic, after which they were rescheduled till January 2021. Furthermore, our study confirms compliance with WHO's [4] recommendation on the use of telemedicine in other to bridge the health access gap caused by the pandemic. Hence, the physical presentation of patients during the pandemic was only encouraged when necessary. However, the study also revealed that patients who attend the diabetes clinic have been adhering to the COVID-19 precautionary protocols such as wearing masks and social distancing most especially when they attend the clinic since it reopened.

Contrary to the assertion made by Ahmed [5] that most of the public hospitals have been converted to COVID-19 treatment centers, which made lots of patients with comorbidities stranded when they needed medical attention, this study found that the University College Hospital diabetes clinic was only closed during the early period of the pandemic in order to ensure the safety of its staff and patients. In addition, the study revealed that the patients were not left stranded by the clinic as telehealth was provided as an alternative was provided. Patients also had access to healthcare personnel on an appointment basis after the clinic reopened in January 2021. Interestingly, the study revealed that patients were encouraged by the clinic to monitor their blood glucose daily by keeping records of it in a book. Another measure instituted was reducing the crowd in the outpatient clinic by extending patients' appointment dates. This had the negative effect of limiting access to healthcare personnel during the pandemic. As a result, many patients with diabetes have had their routine screening deferred [11].

The findings of this study revealed that COVID-19 precautionary measures were strictly adhered to in the clinic, and patients have been compliant. This is, therefore, consistent with the recommendations of the WHO that preventive measures such as social distancing, the use of masks, and hand sanitisers should be adhered to [4]. Awareness of the availability of the telehealth service was high among the participants. However, awareness goes beyond knowing that a service exists. It is concerned with understanding and utilizing that knowledge [22].

Contrary to the findings of Hartmann-Boyce [23], which showed that many patients could not utilize the telehealth service because of their inability to afford the equipment needed for the process, findings from the in-depth interviews revealed that most patients did not utilize telehealth because they did not have reasons to use it. However, very few participants who claimed to utilize the service in this study reported that it was helpful.

The religious nature of Nigerian society is evident in the findings of this study, as most of the participants found succor from the worries and fears of the pandemic in the "Supreme Being". This is consistent with the supposition that most patients with diabetes eased their fear during the pandemic by trusting the divine being and seeking supernatural protection from the same [24]. Reliance on the media for sensitization and updates on the pandemic was noted in the study. This affirms the views of Effiong et al. [25] in Nigeria, where the media was utilized to disseminate messages to the masses on how the virus can be spread and how it can be prevented. Some patients coped by being regular on their medications. This finding is consistent with similar studies [26, 27], as it also revealed that some patients with diabetes in University College Hospital were able to cope when the clinic was closed during the early period of the pandemic by using their last prescriptions before the pandemic to procure more drugs.

Furthermore, the use of traditional herbs alongside medications was shown in the study. This is consistent with the findings from the key informant interviews. This confirms that most non-COVID-19 patients relied more on local medications and homemade remedies to cater for their health during the pandemic [28]. Evidence shows that the burden of diabetes management is more on patients with diabetes mellitus in Nigeria, as about 74.5% of the health care expenditure is self-financed by patients while the government provides only 25.5% [29]. This study found that interventions such as subsidisation, availability of drugs, and reduction of consultation fees would help patients cope better during the pandemic. In addition, approval of the resumption of the University College Hospital diabetes association meeting, shortening waiting time in clinics, creation of more call lines, polite treatment of patients, increased sensitization, and the provision of hygienic toilet facilities would help alleviate the stress of the pandemic on diabetes patients in University College Hospital.

## Strengths and limitations

The study's main strength is that it examined the perception and coping mechanisms of patients with diabetes mellitus in a tertiary health center during the COVID-19 pandemic. The qualitative nature of the study only permitted a small number of participants in the hospital to be interviewed; therefore, it is important to evaluate our findings carefully before extrapolating them to the entire country.

## Implications for public health research

Based on the study's findings, it is imperative for health professionals to routinely conduct psychological assessments for diabetes mellitus patients. Also, the health service managers can design a guidance and assistance programme for patients with diabetes mellitus intended to improve their ability to adopt coping mechanisms.

## Conclusion

The study has shown that patients with diabetes mellitus were not fearful of the COVID-19 pandemic despite their status as diabetes mellitus patients. Diabetes mellitus patients were found to be adherent to the COVID-19 precautionary protocols. The health systems' coping mechanisms to avert the pandemic's negative implications were telehealth, encouragement of daily monitoring of sugar levels, and the provision of avenues for a medication review. Additionally, the patients relied on mass media advice and adherence to safety protocols to cope with COVID-19. Based on the study's outcomes, the government and other healthcare stakeholders must reinforce the resilience of diabetes mellitus patients by alleviating their health burdens during the pandemic. This could be done by subsidizing the prices of drugs, tests, and consultation fees, improving the waiting and appointment system in the clinic, creating an online presence for the University College Hospital Diabetes Association Office, and employing more health personnel in the clinic.

## Supporting information

**S1 Checklist. Strengthening the reporting of observational studies in epidemiology statement checklist.**
(DOCX)

**S1 File. Study questionnaire.**
(PDF)

## Acknowledgments

The authors express their appreciation to the University College Hospital, Ibadan, for the privilege of conducting the study within its diabetes clinic. We also thank everyone for their various contributions and assistance during the study.

## Author Contributions

**Conceptualization:** Olajumoke Ololade Tunji-Adepoju, Obasanjo Afolabi Bolarinwa.

**Data curation:** Olajumoke Ololade Tunji-Adepoju, Obasanjo Afolabi Bolarinwa.

**Formal analysis:** Olajumoke Ololade Tunji-Adepoju, Obasanjo Afolabi Bolarinwa.

**Funding acquisition:** Olajumoke Ololade Tunji-Adepoju.

**Investigation:** Olajumoke Ololade Tunji-Adepoju, Obasanjo Afolabi Bolarinwa.

**Methodology:** Olajumoke Ololade Tunji-Adepoju, Obasanjo Afolabi Bolarinwa.

**Software:** Olajumoke Ololade Tunji-Adepoju.

**Supervision:** Obasanjo Afolabi Bolarinwa.

**Validation:** Olajumoke Ololade Tunji-Adepoju.

**Writing – original draft:** Olajumoke Ololade Tunji-Adepoju, Obasanjo Afolabi Bolarinwa, Richard Gyan Aboagye, Williams O. Balogun.

**Writing – review & editing:** Olajumoke Ololade Tunji-Adepoju, Obasanjo Afolabi Bolarinwa, Richard Gyan Aboagye, Williams O. Balogun.

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
