## [Decision Letter · Decision Letter 0]

25 Sep 2023

PONE-D-23-18592Perception and coping mechanisms among diabetes mellitus patients at the University College Hospital, Ibadan, Nigeria, during the COVID-19 pandemicPLOS ONE

Dear Dr. Bolarinwa,

Thank you for submitting your manuscript to PLOS ONE. After careful consideration, we feel that it has merit but does not fully meet PLOS ONE’s publication criteria as it currently stands. Therefore, we invite you to submit a revised version of the manuscript that addresses the points raised during the review process.

We look forward to receiving your revised manuscript.

Kind regards,

Livhuwani Muthelo

Academic Editor

PLOS ONE

Journal Requirements:

Additional Editor Comments:

Dear Authour

Thank you for submitting the paper titled: Perception and coping mechanisms among diabetes mellitus patients at the University College Hospital, Ibadan, Nigeria, during the COVID-19 pandemic to Plos one Journal

We are pleased to let you know that your manuscript has now passed through the review stage and is ready for major revision. Many manuscripts require a round of revisions, so this is a normal but important stage of the editorial process. The manuscript has the potentialto be published but may not be accepted if the authors do not address substantive issues

REVIEWER 1:

I reviewed an article entitled “Perception and coping mechanisms among diabetes mellitus patients at the University College Hospital, Ibadan, Nigeria, during the COVID-19 pandemic”. It was interesting to read. A few changes were required to improve the general structure of the work.

The abstract is relatively long and may exceed the maximum number of words permitted. It might be enhanced by condensing the introductory and conclusion sections. The results should clearly state the themes and subthemes that emerged from the investigation in the first place and please provide a brief explanation for each subtheme.

Overall, you have justified your research nicely in the introduction. However, the method needs to be modified so that it answers the research's objectives and title.

The study's title clearly states that the research mostly featured Dm patients, and you are interested in learning about their perceptions and coping mechanisms while dealing with the management of DM and pandemic covid-19. However, you did include two professional healthcare workers as respondents in your method. Please explain why they were added.

How did you select participants for the qualitative study? Please give a citation for your response.

How did you assure the respondents' diversity?

Please omit the word “only” in your description of study criteria. Restructure your statement to make the grammar more readable.Restructure your statement to make the grammar more readable.

Who developed the questions for data collection? How do the questions appear? Please give an example of the questions that were asked. Who verifies each of the questions before they are used? How many times did the verification procedure occur?

What if the appointment was cancelled or the participant was unable to keep to the schedule because the interview was held in a hospital setting? How many times did the interview take place in total? How did you conduct the interview? Did you use a local language or English? Please provide specifics.

In terms of data analysis, please describe in detail the thematic process that you used in the study. Who checked the interview verbatim and was it shown to the respondents before it was processed further?

The four requirements to grow trustworthiness and establish rigors were credibility (internal validity), transferability (external validity/generalizability), dependability (reliability), and confirmability (objectivity), how did you confirm all of these aspects?

Because the involvement of two healthcare personnel did not connect well with the title of the study, the study's results require significant restructure. You may incorporate their feedback into another research project. As a result, all of the themes and subthemes must be readjusted.

Please offer specific demographic information about respondents using a qualitative method. This is necessary to see the variability of the individuals selected for the study.

I will go over the discussion after I have the amended version of the work, especially if the results have been corrected.

REVIEWER 2

This is an important study. The quality of the manuscript can be improved Several grammatical and spelling errors noted. Inconsistencies in the use of DM patients/ diabetes patients and sampling method noted. The recruitment, inclusion criteria for the health care professionals is not included. The findings section needs to be reworked. It is lengthy with repetition and no flow. Reporting the healthcare professionals and the patients' quotes under one theme without a clear description together with lack of introductory sentences before direct quotes distorted the findings section. Please revise. Please consider rewording the conclusion to make the statements shorter, clearer and more impactful to the reader. Kindly see an annotated copy with the comments and suggestions.

Reviewers' comments:

Reviewer's Responses to Questions

**Comments to the Author**

1. Is the manuscript technically sound, and do the data support the conclusions?

Reviewer #1: Yes

Reviewer #2: Yes

2. Has the statistical analysis been performed appropriately and rigorously? 

Reviewer #1: No

Reviewer #2: N/A

3. Have the authors made all data underlying the findings in their manuscript fully available?

Reviewer #1: Yes

Reviewer #2: Yes

4. Is the manuscript presented in an intelligible fashion and written in standard English?

Reviewer #1: No

Reviewer #2: No

5. Review Comments to the Author

Reviewer #1: I reviewed an article entitled “Perception and coping mechanisms among diabetes mellitus patients at the University College Hospital, Ibadan, Nigeria, during the COVID-19 pandemic”. It was interesting to read. A few changes were required to improve the general structure of the work.

The abstract is relatively long and may exceed the maximum number of words permitted. It might be enhanced by condensing the introductory and conclusion sections. The results should clearly state the themes and subthemes that emerged from the investigation in the first place and please provide a brief explanation for each subtheme.

Overall, you have justified your research nicely in the introduction. However, the method needs to be modified so that it answers the research's objectives and title.

The study's title clearly states that the research mostly featured Dm patients, and you are interested in learning about their perceptions and coping mechanisms while dealing with the management of DM and pandemic covid-19. However, you did include two professional healthcare workers as respondents in your method. Please explain why they were added.

How did you select participants for the qualitative study? Please give a citation for your response.

How did you assure the respondents' diversity?

Please omit the word “only” in your description of study criteria. Restructure your statement to make the grammar more readable.Restructure your statement to make the grammar more readable.

Who developed the questions for data collection? How do the questions appear? Please give an example of the questions that were asked. Who verifies each of the questions before they are used? How many times did the verification procedure occur?

What if the appointment was cancelled or the participant was unable to keep to the schedule because the interview was held in a hospital setting? How many times did the interview take place in total? How did you conduct the interview? Did you use a local language or English? Please provide specifics.

In terms of data analysis, please describe in detail the thematic process that you used in the study. Who checked the interview verbatim and was it shown to the respondents before it was processed further?

The four requirements to grow trustworthiness and establish rigors were credibility (internal validity), transferability (external validity/generalizability), dependability (reliability), and confirmability (objectivity), how did you confirm all of these aspects?

Because the involvement of two healthcare personnel did not connect well with the title of the study, the study's results require significant restructure. You may incorporate their feedback into another research project. As a result, all of the themes and subthemes must be readjusted.

Please offer specific demographic information about respondents using a qualitative method. This is necessary to see the variability of the individuals selected for the study.

I will go over the discussion after I have the amended version of the work, especially if the results have been corrected.

Reviewer #2: This is an important study. The quality of the manuscript can be improved Several grammatical and spelling errors noted. Inconsistencies in the use of DM patients/ diabetes patients and sampling method noted. The recruitment, inclusion criteria for the health care professionals is not included. The findings section needs to be reworked. It is lengthy with repetition and no flow. Reporting the healthcare professionals and the patients' quotes under one theme without a clear description together with lack of introductory sentences before direct quotes distorted the findings section. Please revise. Please consider rewording the conclusion to make the statements shorter, clearer and more impactful to the reader. Kindly see an annotated copy with the comments and suggestions.

6. PLOS authors have the option to publish the peer review history of their article (what does this mean?). If published, this will include your full peer review and any attached files.

Reviewer #1: No

Reviewer #2: No

---

## [Author Response · Author response to Decision Letter 0]

6 Feb 2024

Response to Editor’s/Reviewers’ Comments

Editor’s comments

Reply

This format is duly followed.

Reply 

The questionnaire has been filled out and included as supporting information.

a) If there are ethical or legal restrictions on sharing a de-identified data set, please explain them in detail (e.g., data contain potentially sensitive information, data are owned by a third-party organisation, etc.) and who has imposed them (e.g., an ethics committee). Please also provide contact information for a data access committee, ethics committee, or other institutional body to which data requests may be sent.

Reply

This is a qualitative study conducted in a small community. We are very careful of realising the recorded voice online to avoid data breaches of someone recognising the participant's voice. As such, releasing it to a third-party website might leave the participants vulnerable to such. However, if it’s provided upon request, the individual/institution making the request can be requested to sign an undertaking not to release the participant's voice in any repository. 

b) If there are no restrictions, please upload the minimal anonymised data set necessary to replicate your study findings as either Supporting Information files or to a stable, public repository and provide us with the relevant URLs, DOIs, or accession numbers. For a list of acceptable repositories, please see http://journals.plos.org/plosone/s/data-availability#loc-recommended-repositories.

Reply

This is a qualitative research design. More information regarding this has been provided above.

4. Please upload a Response to Reviewers letter which should include a point by point response to each of the points made by the Editor and / or Reviewers. (This should be uploaded as a 'Response to Reviewers' file type.) Please follow this link for more information: http://blogs.PLOS.org/everyone/2011/05/10/how-to-submit-your-revised-manuscript/

Reply

This has been done. 

REVIEWER 1:

I reviewed an article entitled “Perception and coping mechanisms among diabetes mellitus patients at the University College Hospital, Ibadan, Nigeria, during the COVID-19 pandemic”. It was interesting to read. A few changes were required to improve the general structure of the work.

Response: Thank you.

The abstract is relatively long and may exceed the maximum number of words permitted. It might be enhanced by condensing the introductory and conclusion sections. The results should clearly state the themes and subthemes that emerged from the investigation in the first place and please provide a brief explanation for each subtheme.

Overall, you have justified your research nicely in the introduction. However, the method needs to be modified so that it answers the research's objectives and title.

Response: Thank you. We have made few revision to the abstract section. However, the abstract is within the range set by PLoS ONE.

The study's title clearly states that the research mostly featured Dm patients, and you are interested in learning about their perceptions and coping mechanisms while dealing with the management of DM and pandemic covid-19. However, you did include two professional healthcare workers as respondents in your method. Please explain why they were added.

How did you select participants for the qualitative study? Please give a citation for your response.

How did you assure the respondents' diversity?

Response: We have addressed these comments.

Please omit the word “only” in your description of study criteria. Restructure your statement to make the grammar more readable. Restructure your statement to make the grammar more readable.

Response: We have removed “only” from the study area.

Who developed the questions for data collection? How do the questions appear? Please give an example of the questions that were asked. Who verifies each of the questions before they are used? How many times did the verification procedure occur? 

Response: The researchers (authors) developed the questionnaires, and the ethnographers assessed its ability to answer the objectives of the study. Pre-testing was done prior to the actual data collection by the researchers in a different facility not used for the data collection.

What if the appointment was cancelled or the participant was unable to keep to the schedule because the interview was held in a hospital setting? How many times did the interview take place in total? How did you conduct the interview? Did you use a local language or English? Please provide specifics.

Response: The interview spanned for at least more than a week, and the participants were informed about this exercise. Hence, the majority of the participants turned up for the interview.

In terms of data analysis, please describe in detail the thematic process that you used in the study. Who checked the interview verbatim and was it shown to the respondents before it was processed further?

Response: We have revised the analysis section.

The four requirements to grow trustworthiness and establish rigors were credibility (internal validity), transferability (external validity/generalizability), dependability (reliability), and confirmability (objectivity), how did you confirm all of these aspects?

Response: We have provided a section under the analysis section describing these four requirements.

Because the involvement of two healthcare personnel did not connect well with the title of the study, the study's results require significant restructure. You may incorporate their feedback into another research project. As a result, all of the themes and subthemes must be readjusted.

Response: The responses have been adjusted to include only in-depth interviews conducted with the DM patients.

Please offer specific demographic information about respondents using a qualitative method. This is necessary to see the variability of the individuals selected for the study.

Response: The authors have addressed this.

I will go over the discussion after I have the amended version of the work, especially if the results have been corrected.

Response: Thank you.

REVIEWER 2

This is an important study. The quality of the manuscript can be improved Several grammatical and spelling errors noted. Inconsistencies in the use of DM patients/ diabetes patients and sampling method noted. The recruitment, inclusion criteria for the health care professionals is not included. The findings section needs to be reworked. It is lengthy with repetition and no flow. Reporting the healthcare professionals and the patients' quotes under one theme without a clear description together with lack of introductory sentences before direct quotes distorted the findings section. Please revise. Please consider rewording the conclusion to make the statements shorter, clearer and more impactful to the reader. Kindly see an annotated copy with the comments and suggestions.

Response: We have revised the manuscript per the comments.

---

## [Decision Letter · Decision Letter 1]

25 Jun 2024

PONE-D-23-18592R1Perception and coping mechanisms of patients with diabetes mellitus during the COVID-19 pandemic in Ibadan, NigeriaPLOS ONE

Dear Dr. Bolarinwa,

Thank you for submitting your manuscript to PLOS ONE. After careful consideration, we feel that it has merit but does not fully meet PLOS ONE’s publication criteria as it currently stands. Therefore, we invite you to submit a revised version of the manuscript that addresses the points raised during the review process.

The Revised manuscript has been re-reviewed by two reviewers and their comments are available below. Reviewer 2 has requested additional details on the recruitment strategies, interview guides and the roles of the researchers to be reported in the methodology. 

We look forward to receiving your revised manuscript.

Kind regards,

Emma Campbell, Ph.D

Staff Editor

PLOS ONE

Reviewers' comments:

Reviewer's Responses to Questions

**Comments to the Author**

1. If the authors have adequately addressed your comments raised in a previous round of review and you feel that this manuscript is now acceptable for publication, you may indicate that here to bypass the “Comments to the Author” section, enter your conflict of interest statement in the “Confidential to Editor” section, and submit your "Accept" recommendation.

Reviewer #1: All comments have been addressed

Reviewer #2: All comments have been addressed

2. Is the manuscript technically sound, and do the data support the conclusions?

Reviewer #1: Yes

Reviewer #2: Yes

3. Has the statistical analysis been performed appropriately and rigorously? 

Reviewer #1: Yes

Reviewer #2: N/A

4. Have the authors made all data underlying the findings in their manuscript fully available?

Reviewer #1: Yes

Reviewer #2: No

5. Is the manuscript presented in an intelligible fashion and written in standard English?

Reviewer #1: Yes

Reviewer #2: Yes

6. Review Comments to the Author

Reviewer #1: I am satisfied with the corrections made. The manuscript looks better and is easier to comprehend. Thank you so much for taking all the comments into consideration. Well done!

Reviewer #2: Methods= study design: reference this information.

Data collection: It is not clear how the participants were recruited. please elaborate. Was there compensation associated with participation in the study? The roles of the researcher/s and research assistants are not clearly outlined. There 4 research assistants, so what were their roles? When was the interview guide pretested? On how many participants? What were the findings of the pretest? Were they included in the main study?

Data analysis: Repetition noted, please proofread and revise. Reference information on thematic data analysis.

Rigour and trustworthiness: The information stated under rigour contradicts a statement made earlier that research assistants were trained anthropologist. Please ensure consistency.

Demographic data and characteristics: Line 237-239: Include the duration in years to avoid confusion and inconsistency. Repetition noted.

Line 362, 392, 472 and 479: typo= please correct.

Include an introductory sentence under telehealth.

Table 2: write the themes in full

7. PLOS authors have the option to publish the peer review history of their article (what does this mean?). If published, this will include your full peer review and any attached files.

Reviewer #1: No

Reviewer #2: No

---

## [Author Response · Author response to Decision Letter 1]

27 Jun 2024

Reviewers' comments and replies

Reviewer #1:

S/N Comments Replies

1 I am satisfied with the corrections made. The manuscript looks better and is easier to comprehend. Thank you so much for taking all the comments into consideration. Well done!

 Many thanks for taking the time to review our manuscript.

Reviewer #2: 

S/N Comments Replies

1 Methods= study design: reference this information.

Data collection: It is not clear how the participants were recruited. please elaborate. Was there compensation associated with participation in the study? The roles of the researcher/s and research assistants are not clearly outlined. There 4 research assistants, so what were their roles? When was the interview guide pretested? On how many participants? What were the findings of the pretest? Were they included in the main study? • In-text citations have been included in the study design. Page 5, lines 121 to 125.

• More information has been added on how the participants were recruited, and interviewed and how much honorarium gift was given to each participant. Page 7, lines 162 to 169.

• The role of the research assistants has been added. Page 7, lines 181 to 182. 

• Information about the pretest has been added on page 8, lines 193 to 196.

2 Data analysis: Repetition noted, please proofread and revise. Reference information on thematic data analysis. All the repetition has been deleted and all sentences have been cited accordingly. Page 7/8, lines 183 to 192.

3 Rigour and trustworthiness: The information stated under rigour contradicts a statement made earlier that research assistants were trained anthropologists. Please ensure consistency. The last information has been replaced with a suitable sentence. Page 9, lines 211 to 212.

4 Demographic data and characteristics: Line 237-239: Include the duration in years to avoid confusion and inconsistency. Repetition noted. All the demographic data and characteristic sections have been rewritten. Page 9, lines 233 to 247.

5 Line 362, 392, 472 and 479: typo= please correct. • Line 362 has been corrected now line 388 on page 15. 

• Line 392 has been corrected now line 419 on page 16.

• Line 472 has been corrected now line 497 on page 20.

• Line 479 has been corrected now line 507 to 508 on page 21.

6 Include an introductory sentence under telehealth. An introductory sentence has been included. Page 16, line 398. 

7 Table 2: write the themes in full The themes have been written in full in table 2.

---

## [Decision Letter · Decision Letter 2]

13 Aug 2024

Perception and coping mechanisms of patients with diabetes mellitus during the COVID-19 pandemic in Ibadan, Nigeria

PONE-D-23-18592R2

Dear Dr. Bolanrinwa

We’re pleased to inform you that your manuscript has been judged scientifically suitable for publication and will be formally accepted for publication once it meets all outstanding technical requirements.

Kind regards,

Sylla Thiam, M.D, MPH

Academic Editor

PLOS ONE

Additional Editor Comments (optional):

Reviewers' comments:

Reviewer's Responses to Questions

**Comments to the Author**

1. If the authors have adequately addressed your comments raised in a previous round of review and you feel that this manuscript is now acceptable for publication, you may indicate that here to bypass the “Comments to the Author” section, enter your conflict of interest statement in the “Confidential to Editor” section, and submit your "Accept" recommendation.

Reviewer #1: All comments have been addressed

Reviewer #2: All comments have been addressed

2. Is the manuscript technically sound, and do the data support the conclusions?

Reviewer #1: Yes

Reviewer #2: (No Response)

3. Has the statistical analysis been performed appropriately and rigorously? 

Reviewer #1: Yes

Reviewer #2: (No Response)

4. Have the authors made all data underlying the findings in their manuscript fully available?

Reviewer #1: Yes

Reviewer #2: Yes

5. Is the manuscript presented in an intelligible fashion and written in standard English?

Reviewer #1: Yes

Reviewer #2: (No Response)

6. Review Comments to the Author

Reviewer #1: I have made my decision to accept this work and will not change it. The manuscript was well written after the revisions, and all concerns have been addressed appropriately. Well done.

Reviewer #2: (No Response)

7. PLOS authors have the option to publish the peer review history of their article (what does this mean?). If published, this will include your full peer review and any attached files.

Reviewer #1: No

Reviewer #2: No

---

## [Editor Report · Acceptance letter]

16 Aug 2024

PONE-D-23-18592R2 

PLOS ONE

Dear Dr. Afolabi Bolarinwa, 

I'm pleased to inform you that your manuscript has been deemed suitable for publication in PLOS ONE. Congratulations! Your manuscript is now being handed over to our production team.

Kind regards, 

on behalf of

Dr. Sylla Thiam 

Academic Editor

PLOS ONE